# Behavioral Emotion Regulation Strategies and Symptoms of Psychological Distress Among Turkish University Students

**DOI:** 10.3390/bs15010006

**Published:** 2024-12-26

**Authors:** Edib Şevki Keskiner, Ertuğrul Şahin, Nursel Topkaya, Zehra Yiğit

**Affiliations:** 1Department of Counseling and Higher Education, Patton College of Education, Ohio University, 1 Ohio University, Athens, OH 45701, USA; 2Department of Guidance and Psychological Counseling, Faculty of Education, Amasya University, Amasya Merkez 05100, Türkiye; ertugrulsahin@amasya.edu.tr; 3Department of Guidance and Psychological Counseling, Faculty of Education, Çanakkale Onsekiz Mart University, Çanakkale 17000, Türkiye; nursel.topkaya@comu.edu.tr; 4Independent Researcher, İstanbul 34000, Türkiye; psyzehrayigit@gmail.com

**Keywords:** behavioral emotion regulation strategies, depression, anxiety, stress, university students, Türkiye

## Abstract

The purpose of this study was to examine the association between behavioral emotion regulation strategies and symptoms of depression, anxiety, and stress among Turkish university students. Participants consisted of 633 students continuing their university education in two different universities in Türkiye. Participants completed a data collection tool comprising a Sociodemographic Information Form, the Behavioral Emotion Regulation Questionnaire, and the Depression Anxiety Stress Scales-21. Data were analyzed using descriptive statistics, Pearson’s product-moment correlation coefficient analysis, and multivariate multiple regression analysis. The results of this study revealed that seeking distraction was negatively associated with symptoms of depression, anxiety, and stress, whereas withdrawal, seeking social support, and ignoring were positively associated with symptoms of depression, anxiety, and stress among university students. Additionally, actively approaching was negatively associated with depressive symptoms. Overall, the findings demonstrate that university students who use maladaptive behavioral emotion regulation strategies (e.g., withdrawal, ignoring) tend to have higher levels of psychological distress, whereas university students who use adaptive emotion regulation strategies (e.g., distraction) tend to have lower levels of psychological distress. However, contrary to expectations, seeking social support was positively associated with symptoms of psychological distress. Given the paucity of research on the relationship between behavioral emotion regulation strategies and psychological distress in the Turkish cultural context, this study may contribute to identifying both universal and culturally specific strategies associated with depressive, anxiety, and stress symptoms among Turkish university students.

## 1. Introduction

The mental health of university students has drawn increasing attention in recent years due to its theoretical significance and practical implications. Research has consistently demonstrated that mental health problems are prevalent among university students ([20]; [48]). Although prevalence estimates vary across studies, mood disorders, anxiety disorders, and substance use disorders are particularly common ([4]; [48]). Recent international studies indicate that approximately one in every two university students exhibits moderate to severe levels of depression, anxiety, and stress symptoms ([41]; [44]; [45]). Similar findings have been reported in Türkiye, where symptoms of depression, anxiety, and stress are common among university students ([5]; [10]).

Research has consistently demonstrated that university students encounter diverse stressors throughout their academic careers, encompassing academic, emotional, social, and economic domains ([7]; [20]; [35]; [44]; [45]). Academic stressors include rigorous coursework, exam pressure, and maintaining satisfactory grade point averages ([21]). Emotional challenges frequently manifest as homesickness, identity exploration, and adaptation to new environments ([7]). Social stressors typically involve navigating new relationships, managing peer pressure, and balancing social activities with academic responsibilities ([35]). Economic stressors commonly stem from tuition costs, living expenses, and student loan debt ([2]). The COVID-19 pandemic has further intensified these challenges and introduced additional complexities to the mental health of students. Factors such as remote learning, social isolation, and uncertainty have intensified the existing difficulties and created novel stress sources for this population ([50]). These stressors can significantly impact students’ mental well-being, potentially triggering latent predispositions or precipitating new psychological problems ([20]; [48]). Moreover, university students must navigate various developmental tasks, including self-discovery, establishing intimate relationships, making career decisions, and achieving independence during this period ([3]). When students struggle to effectively manage these developmental challenges and stressors, they may experience psychological distress manifested as anxiety, stress, and/or depressive symptoms.

Higher levels of depression, anxiety, and stress among university students can lead to several adverse outcomes. These mental health issues can significantly impair academic performance, resulting in decreased grades and increased absenteeism due to impaired concentration, diminished motivation, and reduced energy levels ([21]). Higher levels of psychological distress can also damage relationships with peers and family members, potentially leading to social isolation and feelings of loneliness among college students ([19]; [33]). Moreover, it can negatively affect physical health, resulting in sleep disturbance, nutritional deficiency, and a weakened immune system ([18]). In the long term, untreated symptoms of depression, anxiety, and stress can lead to more serious problems, such as substance use disorders, and suicidal thoughts and attempts ([26]). From an economic perspective, higher levels of psychological distress can extend graduation timelines or lead to attrition, negatively impacting future career prospects and earning potential ([34]). Therefore, supporting and improving the mental health of university students is vital for their individual well-being and social contributions. In this context, identifying variables associated with elevated depression, anxiety, and stress symptoms among university students can facilitate the early detection of at-risk individuals and inform the development of effective prevention and intervention strategies. This approach may mitigate the negative effects of high psychological distress on university students, potentially improving their academic outcomes and promoting positive long-term mental health trajectories.

Emotion regulation strategies may play a significant role in the development and maintenance of depression, anxiety, and stress among university students. Emotion regulation refers to the internal and external processes through which individuals manage their emotional responses ([29]). These processes involve the monitoring, evaluation, and modification of emotional reactions, particularly in terms of their intensity and temporal characteristics ([30]). The concept of emotion regulation is multifaceted, encompassing both implicit cognitive processes, such as reappraisal and rumination, and explicit behavioral processes, like seeking social support and problem-solving ([28]; [37]). The concept of emotion regulation is closely related to coping. While the emergence of emotions may be triggered by a stressor, a stressor is not a necessary condition for emotions to arise. Emotion regulation focuses on managing emotions that occur in response to a stressor or other external factors, whereas coping is specifically employed in response to a stressor. Thus, from a theoretical perspective, emotion regulation is intricately linked to coping ([14]). Emotion regulation strategies are specific techniques or approaches that individuals employ to influence which emotions they experience, when they experience them, and how they experience and express these emotions ([29], [30]). The primary aim of these strategies is to modulate the intensity, duration, or quality of emotional responses to better align with personal goals and situational demands. This modulation process is particularly relevant in the context of university students who face numerous academic, social, and personal challenges that can elicit strong emotional responses.

Effective emotion regulation is a crucial component of optimal functioning and healthy stress adaptation ([30]). This capacity plays a pivotal role in mental health and psychological well-being, influencing how individuals respond to and manage the emotional challenges they encounter in daily life ([14]; [29]; [40]). The ability to modulate emotional responses appropriately is particularly important when facing stressful situations because it allows for more adaptive coping strategies and better overall mental health outcomes ([27]; [29]; [37]). Conversely, difficulties in regulating intense emotional responses to stressful events are considered fundamental mechanisms underlying the relationship between emotions and psychopathology ([1]; [14]). When individuals struggle to manage their emotional reactions effectively, they may experience prolonged or intensified negative emotions, potentially leading to the development or exacerbation of mental health difficulties. A substantial body of research has established a strong link between emotion regulation difficulties and a wide range of mental health disorders, including mood disorders, anxiety disorders, eating disorders, substance use disorders, and personality disorders ([1]; [14]; [40]). Therefore, the development and empowerment of effective emotion regulation skills represent crucial areas for intervention to protect mental health and prevent the development of psychopathology across diverse populations.

[29]’s ([29], [30]) emotion regulation process model provides a comprehensive framework for understanding the strategies individuals employ to manage their emotions in daily life. This model conceptualizes emotion regulation as a dynamic, multistage process that unfolds over time, encompassing five main strategies: situation selection, situation modification, attentional deployment, cognitive change, and response modulation. Situation selection (e.g., behavioral disengagement) and modification (e.g., seeking social support) are proactive approaches that occur early in the emotion-generative process. These strategies involve choosing or altering circumstances to influence emotional outcomes before they fully develop. Attentional deployment, such as distraction, involves redirecting focus and requires less cognitive complexity than subsequent strategies. Cognitive change, such as reappraisal, involves more sophisticated mental processes that alter the meaning or significance of an emotional stimulus. Response modulation, the final stage, encompasses strategies activated after an emotional response occurs, such as the suppression or expression of emotions ([29]). According to this model, emotion regulation is cyclical, with each stage influencing subsequent regulatory efforts. Behavioral actions can serve regulatory purposes at various stages. For example, individuals may avoid anxiety-provoking social situations (situation selection) or consume alcohol to reduce sadness (response modulation). This model views emotion regulation attempts as hedonic in nature, focusing on downregulating negative emotions and upregulating positive emotions. However, it is also recognized that the reverse can occur ([29], [30]).

Emotion regulation encompasses a diverse array of strategies that individuals employ to manage negative and positive emotions. These strategies are classified into two broad categories as cognitive and behavioral emotional regulation strategies ([37]). While both categories play crucial roles in emotional management, research has predominantly focused on cognitive strategies and their impact on mental health ([28]; [27]). Because behavioral strategies allow individuals to modulate their emotional responses, they play a critical role in emotion regulation, particularly in stressful situations. According to [37] ([37]), individuals frequently employ a range of behavioral strategies in response to threatening or stressful life events, including seeking distraction, withdrawal, actively approaching, seeking social support, and ignoring. Seeking distraction refers to engaging in alternative activities to distance oneself from the emotions associated with the stressor. Withdrawal is defined as retreating from situations and social contacts to avoid dealing with the stressful event. Actively approaching involves directly addressing the problem that causes stress through purposeful actions intended to resolve the issue. Seeking social support is characterized by sharing emotions with others and actively seeking advice and support to cope with the stressful situation. Ignoring is defined as acting as if nothing has happened, essentially denying or minimizing the stressful event as a coping mechanism ([37]). According to these researchers, seeking distraction, actively approaching, and seeking social support represent adaptive behavioral emotion regulation strategies, whereas withdrawal and ignoring represent maladaptive behavioral emotion regulation strategies.

Research has consistently demonstrated a strong link between emotion regulation and mental health ([1]; [14]). Previous comprehensive reviews and meta-analyses, such as those by [14] ([14]) and [1] ([1]), have provided important insights into the relationships between various emotion regulation strategies and psychopathology. However, these studies have primarily focused on cognitive emotion regulation strategies, examining their associations with symptoms of depression, anxiety, and other mental health problems. Cognitive emotion regulation strategies involve complex mental processes that shape the emotional experience such as reappraisal and rumination ([1]; [28]; [27]). In contrast, behavioral emotion regulation strategies allow individuals to directly modulate their emotional responses, particularly in the context of stressful situations ([37]). However, behavioral emotion regulation strategies have received less attention in the literature compared to cognitive strategies. We used a different approach in this study by investigating the role of behavioral emotion regulation strategies, as conceptualized by [37] ([37]), to provide a more nuanced understanding of how individuals actively manage their negative emotions through specific actions rather than solely relying on cognitive processes.

Few previous studies have examined the relationship between behavioral emotion regulation strategies and negative mental health indicators, such as depression, anxiety, and stress, and these studies have reported inconsistent results ([8]; [23]; [49]; [55]; [58]). While withdrawal and ignoring strategies are generally found to be negatively associated with symptoms of depression, anxiety, and stress ([8]; [49]; [55]; [58]), findings on the relationship between distraction, actively approaching, and seeking social support and psychological stress are less consistent ([8]; [23]; [49]; [55]; [58]). Although some previous studies have suggested that seeking distraction is either unrelated to depression ([8]; [55]), anxiety ([8]; [55]; [58]), or stress ([8]; [23]), other studies have indicated that seeking distraction is negatively associated with these symptoms ([49]). Similarly, although some studies report no association between actively approaching and anxiety ([8]), other studies have found a negative association between actively approaching and symptoms of depression, anxiety, and stress ([23]; [49]). The association between seeking social support and psychological stress is also quite inconsistent. Some previous studies reported no significant relationship between seeking social support and depression ([8]; [55]) or anxiety ([55]; [58]), whereas [49] ([49]) found that seeking social support was positively associated with depression, anxiety, and stress. Conversely, Foroughi et al. ([23]) reported a negative association between seeking social support and symptoms of depression, anxiety, and stress.

Several factors may contribute to the inconsistent findings among studies that examined behavioral emotion regulation strategies and psychological distress. These factors can include cultural contexts, sampling differences, and methodological differences across studies. For instance, seeking social support may be a positive coping strategy in collectivist cultures but less effective in societies emphasizing independence and autonomy. Additionally, individual psychological needs, contextual factors (e.g., the type or severity of the stressor), and environmental stressors may modulate the effectiveness of these strategies across cultures ([9]). Therefore, investigating how these strategies function across diverse sociocultural contexts is crucial.

Methodological limitations in previous studies further complicated the interpretation of the results. Most previous studies relied on bivariate analyses, such as Pearson product–moment correlation coefficient analysis, to examine the relationships between behavioral emotion regulation strategies and the symptoms of depression, anxiety, and stress ([8]; [23]; [49]; [55]; [58]). While these analyses offer valuable insights into the strength and direction of linear relationships between specific strategies and psychological distress, they fail to account for the complex interplay among different behavioral strategies. However, linear multiple regression analysis and its extensions (e.g., multivariate multiple regression analysis) provide several advantages over correlation analysis ([15]; [16]; [31]).

First, linear multiple regression analysis allows researchers to identify the relative importance of each behavioral emotion regulation strategy in predicting psychological distress, rather than simple association. This information can be particularly useful for informing the development of targeted interventions that focus on strengthening the most influential behavioral emotion regulation strategies. Such knowledge can also enable mental health professionals to develop and tailor interventions more effectively, especially when working with university students. Second, linear multiple regression analysis accounts for the potential overlap or shared variance among the different behavioral strategies, providing a clearer picture of the unique contributions of each behavioral strategy. This is critical because behavioral emotion regulation strategies often do not operate in isolation but interact in complex and interdependent ways. Finally, linear multiple regression analysis enables researchers to assess the overall model fit, indicating the collective explanatory power of the behavioral emotion regulation strategies in predicting symptoms of depression, anxiety, and stress ([15]; [16]; [31]).

Turkish cultural context presents a unique combination of cultural, social, and economic factors that make the university student population particularly relevant for studying behavioral emotion regulation strategies and psychological distress. As a collectivist society, Türkiye emphasizes interdependence, familial obligations, and community ties ([36]), shaping how individuals regulate their emotions and cope with stress. For instance, seeking social support, often regarded as an adaptive strategy, may hold increased importance in Türkiye due to the cultural value placed on close interpersonal connections. Conversely, maladaptive strategies such as ignoring or withdrawal may be influenced by societal norms discouraging open emotional expression in certain contexts. [29]’s ([29], [30]) emotion regulation process model conceptualizes emotion regulation as a dynamic process encompassing multiple strategies. This model provides a comprehensive framework for understanding how individuals manage emotional responses, particularly in culturally specific contexts. For example, while adaptive behavioral strategies such as distraction or active problem-solving may help students to alleviate stress, their effectiveness can vary depending on cultural norms and individual psychological needs. In collectivist cultures like Türkiye, strategies involving interpersonal dynamics, such as seeking social support, may be more frequently employed or perceived differently than in individualistic societies ([36]). Furthermore, maladaptive strategies including withdrawal and ignoring may exacerbate psychological distress by hindering adaptive responses to stressors. [29]’s ([29]) theory underscores the importance of context in shaping how these strategies function, providing a robust basis for investigating the interplay between behavioral emotion regulation strategies and psychological distress in Turkish university students. The present study aims to determine the predictive role of behavioral emotion regulation strategies (seeking distraction, withdrawal, actively approaching, seeking social support, and ignoring) in explaining symptoms of depression, anxiety, and stress among Turkish university students.

[29]’s ([29], [30]) emotion regulation theory suggests that while adaptive behavioral emotion regulation strategies allow individuals to effectively modulate their emotional responses during stressful situations, maladaptive behavioral emotion regulation strategies can lead to psychological distress. While previous findings have been inconsistent, particularly regarding adaptive strategies ([8]; [23]; [49]; [55]; [58]), we expect that in the Turkish cultural context, both adaptive and maladaptive strategies will demonstrate significant associations with psychological distress, with maladaptive strategies showing stronger predictive power (e.g., withdrawal) due to their more direct impact on emotional avoidance and dysregulation ([1]). Based on [29]’s ([29], [30]) emotion regulation process model and previous empirical evidence on behavioral emotion regulation strategies ([8]; [23]; [37]; [49]; [55]; [58]), we test the following hypotheses in this study.

**Hypothesis** **1.**
*Seeking distraction will be negatively associated with symptoms of depression, anxiety, and stress among Turkish university students.*


**Hypothesis** **2.**
*Withdrawal will be positively associated with symptoms of depression, anxiety, and stress among Turkish university students.*


**Hypothesis** **3.**
*Actively approaching will be negatively associated with symptoms of depression, anxiety, and stress among Turkish university students.*


**Hypothesis** **4.**
*Seeking social support will be negatively associated with symptoms of depression, anxiety, and stress among Turkish university students.*


**Hypothesis** **5.**
*Ignoring will be positively associated with symptoms of depression, anxiety, and stress among Turkish university students.*


## 2. Methods

### 2.1. Research Design

In this study, we employed a cross-sectional research design to examine the association between behavioral emotion regulation strategies and symptoms of depression, anxiety, and stress among university students. The dependent variables of the study were depression, anxiety, and stress, while the independent variables were seeking distraction, withdrawal, actively approaching, seeking social support, and ignoring.

### 2.2. Participants

We performed a priori power analyses to determine the minimum sample size required for this study. This step is crucial for ensuring that our research has sufficient statistical power to detect meaningful effects if they exist. Power analyses for multiple linear regression models revealed that a minimum sample size of 402 participants was required to detect a low effect size (*R*^2^ = 0.05) with a statistical power of 0.95 and significance level of 0.05 for five predictor variables ([22]). The decision to use a small effect size in power analyses was based on several factors. Small effect sizes can detect small but practically significant differences in the population, enhance the precision of statistical estimates, effectively control Type II error rates, and improve the reproducibility and generalizability of findings ([46]). We aimed to recruit at least 500 participants to account for potentially incomplete questionnaires, missing values, and outliers and to better represent the study population. A total of 633 students from two universities in the Central Black Sea Region and the Marmara Region of Türkiye participated in this study. Participants were recruited using the convenience sampling method.

### 2.3. Measures

#### 2.3.1. Sociodemographic Information Form

Participants completed this form to provide information about their gender, age, and grade level.

#### 2.3.2. Depression Anxiety Stress Scales-21

We assessed depression, anxiety, and stress symptoms using the Depression Anxiety Stress Scales (DASS; [42]) This study used the 21-item short form (DASS-21) of the Depression Anxiety Stress Scales, which also exists as a 42-item long form (DASS-42). Previous studies have provided strong evidence regarding the construct validity, convergent and discriminant validity, and internal consistency of the DASS-21, demonstrating that the scale is a valid and reliable measurement tool for assessing depression, anxiety, and stress symptoms ([23]; [49]; [52]; [57]). [52] ([52]) examined the validity and reliability of the DASS-21 among Turkish adults using translated items from the DASS-42 long form ([10]). The results of this study provided strong evidence for the construct, convergent, and discriminant validity and reliability of the DASS-21; replicated the original scale’s three-factor structure; and confirmed its suitability for Turkish adults. Confirmatory factor analysis (CFA) also indicated that the original three-factor structure of the DASS-21 had a good fit with the data on this sample of university students (χ^2^[186] = 437.60, *p* < 0.001, χ^2^/df = 2.35, Comparative Fit Index [CFI] = 0.92, Tucker–Lewis Index [TLI] = 0.91, Root Mean Square Error of Approximation [RMSEA] = 0.05, *p* > 0.05, RMSEA 90% Confidence Interval Lower Bound = 0.04, RMSEA 90% Confidence Interval Upper Bound = 0.05, Standardized Root Mean Square Residual [SRMR] = 0.04).

The DASS-21 is a self-report instrument comprising three 7-item subscales that measure depression, anxiety, and stress symptoms. The depression subscale assesses the presence and severity of depressive symptoms related to feelings of anhedonia, dysphoria, inertia, worthlessness, lack of meaning in life, and reluctance to participate in daily activities. The anxiety subscale assesses the presence and severity of anxiety symptoms related to the subjective experience of anxious affect, situational anxiety, autonomic arousal, and skeletal muscle effects. Finally, the stress subscale assesses the presence and severity of stress symptoms related to nervous arousal, agitation, irritability, overreactivity, impatience, and difficulty in relaxing ([42]). Respondents rate each item on a 4-point Likert-type scale ranging from 0 (*Did not apply to me at all*) to 3 (*Applied to me very much or most of the time*), taking into account their mood over the past week. Scores for each subscale range from 0 to 21, with higher scores indicating increased symptom severity. In this study, the Cronbach alpha coefficients for the depression, anxiety, and stress subscales were calculated as 0.87, 0.80, and 0.77, respectively, indicating good internal consistency reliability. Sample items include “I felt that life was meaningless” for depression, “I felt I was close to panic” for anxiety, and “I found it difficult to relax” for stress.

#### 2.3.3. Behavioral Emotion Regulation Questionnaire

We measured behavioral emotion regulation strategies using the Behavioral Emotion Regulation Questionnaire (BERQ), developed by [37] ([37]). Previous studies across different cultures and samples have provided strong evidence for the construct, convergent, and divergent validity; internal consistency reliability; and test–retest reliability of the BERQ ([8]; [23]; [49]; [58]). [55] ([55]) conducted the Turkish adaptation, validity, and reliability studies of the BERQ. The results of the validity and reliability studies indicated that the questionnaire is a valid and reliable measurement tool with a five-factor structure similar to the original questionnaire. In this study, the five-factor structure of the BERQ was also confirmed in university students (χ^2^[158] = 447.15, *p* < 0.001, χ^2^/df = 2.83, CFI = 0.90, TLI = 0.88, RMSEA = 0.05, *p* > 0.05, RMSEA 90% Confidence Interval Lower Bound = 0.04, RMSEA 90% Confidence Interval Upper Bound = 0.05, SRMR = 0.08).

The BERQ comprises five subscales. These subscales were seeking distraction, withdrawal, actively approaching, seeking social support, and ignoring. Each subscale contains four items. Respondents rate each item on a 5-point Likert-type scale ranging from 1 ([*almost*] *never*) to 5 ([*almost*] *always*). Scores for each subscale range from 4 to 20, with higher scores indicating a more frequent use of the corresponding behavioral emotion regulation strategy. In a sample of healthy Turkish adults, [55] ([55]) reported Cronbach’s alpha internal consistency reliability coefficients ranging from 0.72 (distraction) to 0.88 (seeking social support). In the present study, Cronbach’s alpha coefficients were calculated as 0.65 for distraction, 0.77 for withdrawal, 0.80 for actively approaching, 0.87 for seeking social support, and 0.76 for ignoring subscales. Sample items include “I do other things to distract myself” for seeking distraction, “I avoid other people” for withdrawal, “I take action to deal with it” for actively approaching, “I ask someone for advice” for seeking social support, and “I move on and pretend that nothing happened” for ignoring.

### 2.4. Procedure

Prior to commencing the study, ethical approval was obtained from the Ethics Review Board of Amasya University. We conducted a pilot study with 15 university students to assess the applicability and comprehensibility of the scales and identify any ambiguous items. All participants reported that the survey items were clear, easy to read, and understandable. Data collection was conducted over a period of approximately three months between March 2024 and June 2024. The data collection tool consisted of a short letter explaining the purpose of the study, an informed consent form, and a copy of the measures. The study was announced to university students through classroom announcements and social media platforms (Instagram, Twitter, Telegram, and Facebook). Interested students were asked to complete the survey using Google Forms, a cloud-based data management tool commonly used for web-based surveys. In accordance with ethical principles, participants were informed about the confidentiality of their data, their right to withdraw from the study at any time without any sanctions, the purpose and importance of the research, and the voluntary nature of the study. All university students provided informed consent and participated in the study voluntarily without any incentives or rewards.

### 2.5. Statistical Analysis

We conducted all statistical analyses using IBM SPSS Statistics (Version 26) and Stata Statistical Software (Version 15). Prior to the main analyses, we performed preliminary data screening to assess data accuracy, missing values, outliers, and statistical assumptions ([31]; [53]). All variables were within expected ranges. Our use of a mandatory response format in Google Forms precluded missing data. We identified univariate outliers using z-scores, while we detected multivariate outliers using Cook’s distance ([31]; [53]). We found no univariate or multivariate outliers in our dataset. We used descriptive statistics, including frequencies, percentages, means, and standard deviations, to provide information about the gender, age, and grade level of the participants. We conducted Pearson product–moment correlation analyses to determine the strength and direction of the linear relationships among depression, anxiety, stress, seeking distraction, withdrawal, actively approaching, seeking social support, and ignoring. Multivariate multiple regression analysis was employed to identify predictors of depression, anxiety, and stress in university students, given its appropriateness for examining interrelated dependent variables ([16]). To assess the robustness of these relationships and control for potential sociodemographic effects, we performed two additional multivariate multiple regression analyses. Due to the strong correlation between grade level and age (*r*_s_ = 0.73), we tested these sociodemographic variables in separate models to avoid multicollinearity. Model 1 included gender, age, and behavioral emotion regulation strategies as predictors, while Model 2 incorporated gender, grade level, and behavioral emotion regulation strategies. These supplementary analyses yielded results consistent with our primary findings, suggesting that the relationships between emotion regulation strategies and symptoms of depression, anxiety, and stress remained stable across sociodemographic variables (see Appendix A). Thus, we present our main findings without sociodemographic covariates.

Multivariate multiple regression analysis allows researchers to collectively examine the relationships between all independent variables and the set of dependent variables, as well as the effect on each dependent variable individually. Although it yields the same beta coefficients, standard errors, *p*-values, and confidence intervals as separate multiple regression analyses for each dependent variable, it offers additional advantages. These include the ability to compare beta coefficients across regression equations and test the overall effect when predicted variables are interrelated ([16]). Prior to the main analysis, we examined and confirmed that our data met the assumptions of normality, linearity, homoscedasticity, and the absence of multicollinearity ([16]; [31]; [53]). We interpreted correlation coefficients and explained variance ratios using [13]’s ([13]) effect size classification. For correlation coefficients, we considered values of 0.29 or below, 0.30 to 0.49, and 0.50 or above as small, medium, and large effect sizes, respectively. For the explained variance (*R*^2^), we interpreted values of 0.12 or below, 0.13 to 0.25, and 0.26 or above as small, medium, and large effect sizes, respectively. We used a significance level of *p* < 0.05 in all inferential statistical analyses.

## 3. Results

### 3.1. Sociodemographic Characteristics of the Participants

Table 1 presents the sociodemographic characteristics of the participants including gender, age, and grade level distribution of the participants.

As shown in Table 1, there were 477 female (75.4%) and 156 male (24.6%) students in the study sample. The age distribution of the participants ranged from 18 to 28 years, with a mean age of 21.28 years (*SD* = 1.77). Regarding academic grade level distribution, fourth-year students constituted the largest group (*n* = 209; 33%), followed by third-year students (*n* = 158; 25%). The remaining participants were distributed between second-year (*n* = 140; 22.1%) and first-year students (*n* = 126; 19.9%), indicating a relatively balanced representation across academic years with a slightly greater representation of upper-level students.

### 3.2. Pearson Product–Moment Correlation Analyses

Table 2 presents descriptive statistics and Pearson product–moment correlation coefficients between symptoms of depression, anxiety, and stress and the behavioral emotion regulation strategies of seeking distraction, withdrawal, actively approaching, seeking social support, and ignoring.

As shown in Table 2, depression scores were weakly and negatively correlated with actively approaching (*r* = −0.28), weakly and positively correlated with ignoring (*r* = 0.18), and moderately positively correlated with withdrawal (*r* = 0.48). Similarly, anxiety scores were weakly and negatively correlated with actively approaching (*r* = −0.15), weakly and positively correlated with ignoring (*r* = 0.15), and moderately positively correlated with withdrawal (*r* = 0.40). Finally, stress scores were weakly and negatively correlated with actively approaching (*r* = −0.15), weakly and positively correlated with ignoring (*r* = 0.15) and seeking social support (*r* = 0.10), and moderately positively correlated with withdrawal (*r* = 0.38). No significant correlations were found between seeking distraction and depression, anxiety, and stress scores. Additionally, seeking social support was not significantly correlated with depression or anxiety scores.

### 3.3. Multivariate Multiple Regression Analysis

We performed a multivariate multiple regression analysis to determine the extent to which behavioral emotion regulation strategies predicted depression, anxiety, and stress scores. The summary statistics for the multivariate multiple regression analysis are presented in Table 3, and results of the regression analysis are provided in Table 4.

The multivariate multiple regression analysis revealed a significant multivariate main effect (*Wilk’s Lambda* = 0.42, *F*[237, 1653.4] = 2.35, *p* < 0.001), indicating that at least one behavioral emotion regulation strategy significantly predicted one or more of the dependent variables (depression, anxiety, or stress). To further examine these relationships, we conducted follow-up univariate multiple regression analyses for each dependent variable.

As shown in Table 3, the multivariate multiple regression analysis revealed that the multiple linear regression models for depression (*F*[5, 627] = 45.15, *p* < 0.001, *R*^2^ = 0.26), anxiety (*F*[5, 627] = 30.57, *p* < 0.001, *R*^2^ = 0.20), and stress (*F*[5, 627] = 28.59, *p* < 0.001, *R*^2^ = 0.19) were significant. While the depression regression model had a large effect size, the anxiety and stress regression models had medium effect sizes. As shown in Table 4, seeking distraction negatively predicted depression, anxiety, and stress scores in university students. Conversely, withdrawal, seeking social support, and ignoring positively predicted these outcomes. Actively approaching negatively predicted depression scores but was not a significant predictor of anxiety or stress.

These results demonstrate that university students who less frequently use seeking distraction and active approaching strategies, while more often employing withdrawal, seeking social support, and ignoring strategies, tend to report higher levels of depressive symptoms. Similarly, students who employed lower levels of distraction and higher levels of withdrawal, seeking social support, and ignoring also reported higher levels of anxiety and stress symptoms.

## 4. Discussion

In this study, we examined the relationship between behavioral emotion regulation strategies and the symptoms of depression, anxiety, and stress in university students. Our findings revealed that students who frequently employed the seeking distraction strategy exhibited lower levels of depression, anxiety, and stress symptoms. These results are consistent with our research hypothesis and previous research conducted across various clinical and non-clinical samples and cultures, which demonstrated that individuals who frequently utilized the distraction emotion regulation strategy tended to report fewer symptoms of depression, anxiety, or stress ([23]; [37]; [49]; [51]). In a mixed sample of clinical and non-clinical adults, [49] ([49]) found that seeking distraction as a behavioral emotion regulation strategy was negatively associated with symptoms of depression, anxiety, and stress. Similarly, [51] ([51]) reported that distraction was negatively associated with depression and anxiety symptoms and positively associated with successful emotion regulation and life satisfaction. Studies examining the short-term and long-term effects of the distraction emotion regulation strategy indicate that when used consciously and appropriately depending on the context and situation, this strategy can yield positive outcomes ([11]; [25]; [38]; [47]; [51]; [56]). These positive outcomes include providing short-term relief from intense negative emotions, preventing rumination, enhancing effective use of cognitive resources, improving problem-solving skills, allowing time for the development of more effective emotion regulation skills, strengthening psychological resilience, increasing subjective well-being, and protecting physical health ([11]; [25]; [38]; [47]; [51]; [56]).

Our findings suggest that distraction may serve as a functional and adaptive behavioral emotion regulation strategy for reducing psychological distress among Turkish university students. This effectiveness may be partly attributed to the collectivist structure and strong social ties characteristic of Turkish culture ([36]). In the Turkish cultural context, distraction strategies often involve participation in social activities or spending time with family and friends. These activities may enhance psychological well-being through multiple mechanisms, including an increased sense of belonging, provision of emotional support in stressful situations, short-term relief from intense negative emotions, and more effective use of cognitive resources. The effectiveness of distraction as a behavioral emotion regulation strategy in reducing psychological distress among Turkish university students may also be related to its alignment with cultural norms and expectations. For example, distracting activities such as spending time with friends or family in stressful situations are common and accepted behaviors in Turkish culture. Therefore, the use of a distraction strategy may not only serve as a short-term relaxation mechanism in stressful situations but also as a reinforcer that strengthens social bonds, helping university students reduce symptoms of depression, anxiety, and stress.

Our findings show that the withdrawal behavioral emotion regulation strategy is positively associated with symptoms of depression, anxiety, and stress among university students. Specifically, students who frequently employed withdrawal strategies reported higher levels of psychological distress symptoms. Moreover, withdrawal was the strongest predictor of symptoms of depression, anxiety, and stress. These results are consistent with our research hypothesis and previous research across diverse samples and cultures, which consistently demonstrated positive correlations between withdrawal and negative mental health outcomes ([8]; [49]; [55]; [58]). For example, [8] ([8]) found that withdrawal was positively correlated with depression, anxiety, and psychological stress symptoms in Indian adults. Similarly, [55] ([55]) found that withdrawal significantly predicted higher levels of depression and anxiety among Turkish university students. Cognitive theories and empirical research on psychological distress posit that withdrawal, conceptualized as a dysfunctional behavioral coping style, may contribute to the development of negative cognitive biases and distortions. In stressful situations, individuals employing withdrawal strategies may experience increased rumination and engage in avoidance behaviors ([6]; [39]; [56]).

Although the withdrawal strategy may offer short-term relief for Turkish university students following distressing events, its frequent use as a coping style may have detrimental long-term effects. [6] ([6]) suggested that persistent avoidance of problems or emotional difficulties may lead to increased feelings of hopelessness and helplessness, difficulty in reframing negative thoughts about stressful situations, and heightened emotional distress. These negative consequences can then lead to worsening symptoms of depression, anxiety, and stress. In Turkish culture, strong family and social ties increase the tendency of people to seek emotional support and social acceptance in stressful situations ([36]). However, withdrawal as a behavioral coping strategy may weaken these social connections and increase perceptions of social isolation ([19]). This situation could lead to increased psychological distress, particularly among university students undergoing developmental transition, as perceived social support diminishes. The inhibition or limited expression of emotions in Turkish culture ([36]) may also contribute to the more frequent adoption of withdrawal as a behavioral coping mechanism among university students. The continuous use of this strategy may limit university students’ capacity to effectively cope with stressors they encounter in both academic and social areas and may contribute to increased symptoms of depression, anxiety, and stress over time.

We found that the behavioral emotion regulation strategy of actively approaching was a significant negative predictor of depression symptoms in university students, but it did not significantly predict anxiety or stress symptoms. These results partly support our research hypothesis. Actively approaching is a problem-focused behavioral coping strategy that involves taking action to eliminate stressful situations or manage emotional reactions to them. [37] ([37]) classified actively approaching as an adaptive emotion regulation mechanism because it involves actively addressing problems rather than avoiding or suppressing emotions. However, when controlling for other behavioral emotion regulation strategies, studies examining the relationship between actively approaching and negative mental health indicators, such as depression, anxiety, and stress, yielded mixed results. While some studies support this classification ([49]), others report divergent findings ([55]; [58]). Our results align with previous research showing that actively approaching was not associated with anxiety ([55]; [58]) or stress ([8]). Additionally, our findings are consistent with [41]’s ([41]) study of Brazilian university students, which demonstrated that active coping significantly predicted depression but not anxiety or stress.

Our findings indicate the differential effect of the behavioral emotion regulation strategy of actively approaching on psychological stress. This differential effect may be due to the varying impacts of emotion regulation strategies on mental health indicators or the distinct characteristics of depression, anxiety, and stress symptoms. Research has demonstrated that different emotion regulation strategies play unique roles in the etiology and maintenance of various psychological disorders ([1]; [14]; [40]). For instance, previous studies have reported that the problem-solving emotion regulation strategy was more strongly negatively associated with depression than other adaptive coping styles such as acceptance and reappraisal ([1]; [40]). Actively approaching stressful problems may be particularly effective in reducing depressive symptoms, which often involve feelings of helplessness and hopelessness. Students who actively confront their stressors may regain a sense of control and develop feelings of mastery, which in turn may help alleviate their depressive symptoms ([12]). Moreover, research indicates that anxiety and stress are often situational and related to immediate or future threats, whereas depression is more frequently past-oriented and linked to a sense of loss or failure ([6]; [41]; [43]). Therefore, actively approaching stressors may not immediately alleviate anxiety or stress, particularly when they are persistent, perceived as uncontrollable, or have uncertain outcomes (e.g., upcoming exams, health concerns, or ongoing social pressures).

We found that seeking social support was a significant positive predictor of symptoms of depression, anxiety, and stress in university students. These results are not consistent with our research hypothesis. This strategy involves turning to interpersonal resources proactively for emotional sharing, help, and guidance when coping with stressful situations ([24]; [54]). Because it engages an individual’s social network as a coping mechanism and intentionally initiates interpersonal interactions to mitigate stressor effects, seeking social support can be an effective strategy for increasing personal resources and reducing feelings of loneliness and isolation. Thus, [37] ([37]) theoretically classified seeking social support as an adaptive emotion regulation mechanism, with some previous research partially supporting this classification ([23]; [58]). However, contrary to this classification, our results and those of previous studies demonstrate that seeking social support is positively associated with higher levels of depression, anxiety, and stress symptoms ([8]; [49]). These divergent findings highlight the complex and contextual nature of seeking social support and indicate potential cultural differences in how emotion regulation strategies affect mental health indicators.

The unexpected positive association between seeking social support and symptoms of depression, anxiety, and stress can be explained from the perspectives of [29]’s ([29], [30]) emotion regulation theory, Lazarus and Folkman’s stress and coping theory ([9]), and the stress-mobilizing hypothesis ([17]) as well as findings from previous studies ([24]; [54]). [29]’s ([29], [30]) emotion regulation theory suggests that seeking social support can be viewed as an antecedent-focused strategy, where individuals proactively attempt to alter the emotional impact of a stressor by engaging others. While this strategy often has adaptive potential, its effectiveness depends on whether the social support aligns with the individual’s needs and expectations. If the support received is mismatched (e.g., emotional support when instrumental help is required) with the recipient needs, or if the interaction between the support recipient and provider intensifies negative emotions, the strategy may fail to regulate distress effectively. Moreover, Lazarus and Folkman’s stress and coping theory ([9]), as well as previous studies ([24]; [54]), highlight the individual and contextual factors that may influence the effectiveness of seeking social support as a coping mechanism, including the relationship between the support recipient and provider, source of support, type of support, perceived versus received support, nature and severity of the stressors, and current level of psychological distress of the support recipients. For example, students experiencing higher levels of psychological distress may have a reduced ability to perceive support as helpful, even if it is offered.

Additionally, the nature and severity of the stressors, combined with the support provider’s characteristics such as emotional availability or empathy, can influence the expected outcome for students. These complex dynamics can create a paradoxical situation where increased support-seeking behaviors may actually function as a stress-mobilizing mechanism ([17]). Specifically, highly distressed students may intensify their attempts to access their social network, but unsuccessful relief efforts can generate a self-perpetuating cycle of unfulfilled needs and escalating distress. In the context of Turkish university students, these theoretical perspectives suggest that increased seeking social support behaviors as a coping mechanism may predict higher levels of psychological distress. However, given the complex interplay of factors involved and inconsistent findings in current literature, further research is necessary to identify the specific mechanisms and contextual conditions in which seeking social support effectively reduces psychological distress in this population.

We found that Turkish university students who frequently used the behavioral emotion regulation strategy of ignoring tended to exhibit higher levels of depression, anxiety, and stress. These findings are consistent with our research hypothesis and previous studies that demonstrated a positive association between ignoring and symptoms of psychological distress ([8]; [37]; [58]). Ignoring is an avoidance-based, maladaptive coping strategy in which individuals attempt to avoid dealing with a stressor by evading the problem itself or escaping the associated emotional distress. Research consistently shows that avoidance-based emotion regulation strategies are positively associated with psychological distress symptoms such as depression, anxiety, and stress ([1]; [14]). Other research results indicate that frequent use of avoidance-based emotion regulation strategies, such as ignoring, may lead individuals to avoid emotional processing of stressful situations, higher levels of rumination, reduced utilization of adaptive emotion regulation strategies, and diminished social support ([14]; [32]; [47]; [54]). Consequently, university students who frequently employ ignoring strategies may experience cumulative emotional distress over time because of ineffective coping with stressful situations. Specifically, avoidance-based strategies may exacerbate psychological distress in the long term by delaying problem resolution. Thus, the tendency to ignore problems may elevate the levels of depression, anxiety, and stress in Turkish university students.

### 4.1. Limitations

This study has several limitations that should be considered when interpreting the results. First, while we recruited a robust sample size of 633 students that exceeded our a priori power analyses requirements, our sample was limited to a convenience sample of university students from two institutions in the Central Black Sea Region and the Marmara Region of Türkiye. This non-probability sampling method restricts the generalizability of our findings to university students in other regions of Türkiye and internationally, thereby limiting its external validity. The specific characteristics of our sample, drawn from these two distinct geographical regions, may not fully represent the Turkish university student population or university students living in other countries. Second, we relied on self-report measures to assess the symptoms of depression, anxiety, and stress as well as behavioral emotion regulation strategies. Although widely used in psychological research, self-report measures are subject to several potential biases and limitations. These include social desirability bias, where participants may respond in ways they perceive as more socially acceptable, and lack of insight into one’s own emotional states and behaviors. Additionally, memory errors may affect the accuracy of reported experiences, and individual differences in the capacity to accurately assess and report on emotional states and behaviors can influence the data. These factors may have affected the accuracy and reliability of our data and should be considered when interpreting the results.

Third, in this study, we measured depression, anxiety, and stress symptoms using the DASS-21 and behavioral emotion regulation strategies using the BERQ. Although DASS-21 is a widely used tool with strong psychometric properties for assessing these symptoms in various populations, it is not a clinical diagnostic instrument. DASS-21 only provides information about the severity of core symptoms of depression, anxiety, and stress within a defined period. Therefore, the severity of depression, anxiety, or stress symptoms identified using the DASS-21 are not indicative of clinical diagnosis. Additionally, although the BERQ provides information about specific behavioral emotion regulation strategies that individuals use in specific situations (e.g., stressful), it does not provide information about general emotion regulation strategies that individuals can employ in different situations and contexts. Consequently, the findings obtained in this study reflect the participants’ behavioral emotion regulation strategies during stressful events or situations but do not fully reflect their general emotion regulation competence. Fourth, we conducted this study using a cross-sectional design, which precludes causal inferences about the relationships between variables. To draw more definitive conclusions about the directionality and causality of these relationships, future researchers should employ longitudinal and experimental research designs. Finally, our study did not include potential mediating or moderating variables, such as personality traits and perceived social support levels, which could influence the relationships we examined. More comprehensive models that incorporate these factors can enhance our understanding of the connections between behavioral emotion regulation strategies and symptoms of depression, anxiety, and stress. To address these limitations, future researchers should employ larger, more diverse samples and multiple data collection methods. We recommend conducting multi-site studies that include university students from other regions of Türkiye as well as cross-cultural studies to test the generalizability of these findings. Additionally, we recommend that future researchers use longitudinal and experimental designs and examine potential mediating or moderating variables.

### 4.2. Practical Implications

The results of this study have important practical implications for researchers, mental health professionals, and policy makers for the early identification of at-risk students, development of evidence-based prevention and intervention programs, and implementation of holistic policies that promote mental health in university students. Emotion regulation strategies are linked to both mental disorders and their underlying causes ([1]). Therefore, identifying university students’ behavioral emotion regulation strategies can be a critical step in assessing those at risk. Given that our research findings indicate that university students who frequently use certain behavioral emotion regulation strategies are more likely to experience psychological distress, we suggest that researchers and mental health professionals should conduct screening studies to assess emotion regulation strategies. These screening studies can help identify students who consistently engage in maladaptive emotion regulation strategies, such as ignoring or withdrawal, while underutilizing adaptive strategies like seeking distraction. The outcomes of these screening studies can be used to identify students at higher risk of depression, anxiety, and stress based on their emotion regulation profiles.

We recommend that mental health professionals incorporate strategies to reduce maladaptive behavioral emotion regulation (e.g., ignoring, withdrawing) and increase adaptive strategies (e.g., distraction) in their treatment plans for university students experiencing higher levels of psychological distress. Mental health professionals can also use cognitive restructuring techniques to identify beliefs that reinforce maladaptive behavioral emotion regulation strategies (e.g., “Ignoring my problems will make them go away”) and help these clients challenge them ([6]). Additionally, they can teach problem-solving skills and assign healthy distraction activities such as physical exercise, journaling, drawing, and breathing exercises as homework exercises to enhance adaptive behavioral emotion regulation strategies in students experiencing higher levels of psychological distress. Furthermore, mental health professionals can organize informational activities such as seminars, workshops, and psychoeducation across campuses to inform students about the effects of adaptive and maladaptive behavioral emotion regulation strategies on mental health and teach them effective emotion regulation strategies. Finally, policymakers should support research initiatives focused on understanding and improving behavioral emotion regulation in young adults to help college students protect their mental health.

## 5. Conclusions

In conclusion, we examined the relationship between behavioral emotion regulation strategies and symptoms of depression, anxiety, and stress among Turkish university students. We found that students who used dysfunctional behavioral emotion regulation strategies (e.g., withdrawal, ignoring) reported higher levels of psychological distress, whereas those who employed functional strategies (e.g., distraction) exhibited lower levels of psychological distress. These findings suggest that adaptive emotion regulation strategies may serve as protective factors for the psychological health of university students, whereas maladaptive strategies may increase their risk of psychological distress. However, contrary to our expectations, the strategy of seeking social support was positively associated with higher levels of psychological distress. This finding highlights the complex nature of seeking social support as a behavioral emotion regulation strategy, indicating that its use may reflect heightened psychological distress among Turkish university students. Given the scarcity of cross-cultural research on the relationship between behavioral emotion regulation strategies and psychological distress, our findings can contribute to identifying both universal and culture-specific behavioral emotion regulation strategies associated with depression, anxiety, and stress symptoms in Turkish university students.

## Figures and Tables

**Table 1 behavsci-15-00006-t001:** Sociodemographic characteristics of the participants.

Variables	Values
**Sex, *n* (%)**	
Male	156 (24.6)
Female	477 (75.4)
**Age**	
*M* [Min., Max.]	21.28 [18, 28]
**Academic grade level, *n* (%)**	
First-year student	126 (19.9)
Second-year student	140 (22.1)
Third-year student	158 (25.0)
Fourth-year student	209 (33.0)

Note. *N* = 633.

**Table 2 behavsci-15-00006-t002:** Results of Pearson product–moment correlation coefficient analyses.

	1	2	3	4	5	6	7	8
1. Depression								
2. Anxiety	0.64 ***							
3. Stress	0.66 ***	0.72 ***						
4. Seeking distraction	−0.06	−0.04	−0.01					
5. Withdrawal	0.48 ***	0.40 ***	0.38 ***	0.03				
6. Actively approaching	−0.28 ***	−0.15 ***	−0.15 ***	0.25 ***	−0.36 ***			
7. Seeking social support	−0.00	0.06	0.10 *	0.23 ***	−0.17 ***	0.30 ***		
8. Ignoring	0.18 ***	0.15 ***	0.15 ***	0.36 ***	0.31 ***	−0.16 ***	−0.16 ***	
*M*	7.15	6.45	7.90	12.45	11.04	12.74	13.21	10.54
*SD*	4.25	3.63	3.52	2.63	3.30	2.89	3.59	3.40
Minimum	0.00	0.00	0.00	5.00	4.00	4.00	4.00	4.00
Maximum	21.00	18.00	21.00	20.00	20.00	20.00	20.00	20.00
Skewness	0.78	0.48	0.54	−0.02	0.19	−0.10	−0.25	0.27
Kurtosis	0.55	0.13	0.51	−0.19	−0.27	−0.01	−0.47	−0.20

Note. * *p* < 0.05, *** *p* < 0.001.

**Table 3 behavsci-15-00006-t003:** Model summary statistics for multiple regression models.

Model	*R*	*R* ^2^	*SE Est.*	*F*	*df* _1_	*df* _2_	*p*
Depression	0.51	0.26	3.66	45.15	5	627	0.001 ***
Anxiety	0.44	0.20	3.27	30.57	5	627	0.001 ***
Stress	0.43	0.19	3.18	28.59	5	627	0.001 ***

Note. *** *p* < 0.001.

**Table 4 behavsci-15-00006-t004:** Results of depression, anxiety, and stress multiple regression analyses.

Model	*B*	*SE*	β	*t*	*p*
**Depression**					
Intercept	1.95	1.19		1.64	0.102
Seeking distraction	−0.17	0.06	−0.11	−2.65	0.008 **
Withdrawal	0.56	0.05	0.43	11.30	0.001 ***
Actively approaching	−0.19	0.06	−0.13	−3.20	0.001 **
Seeking social support	0.18	0.04	0.15	4.04	0.001 ***
Ignoring	0.11	0.05	0.09	2.27	0.024 *
**Anxiety**					
Intercept	0.40	1.07		0.37	0.710
Seeking distraction	−0.17	0.06	−0.12	−2.90	0.004 **
Withdrawal	0.44	0.04	0.40	10.00	0.001 ***
Actively approaching	−0.02	0.05	−0.02	−0.44	0.660
Seeking social support	0.19	0.04	0.18	4.68	0.001 ***
Ignoring	0.11	0.04	0.10	2.38	0.018 *
**Stress**					
Intercept	1.91	1.04		1.84	0.066
Seeking distraction	−0.12	0.06	−0.09	−2.12	0.034 *
Withdrawal	0.40	0.04	0.37	9.28	0.001 ***
Actively approaching	−0.06	0.05	−0.05	−1.11	0.268
Seeking social support	0.21	0.04	0.22	5.48	0.001 ***
Ignoring	0.10	0.04	0.09	2.19	0.029 *

Note. * *p* < 0.05, ** *p* < 0.01, *** *p* < 0.001.

## Data Availability

The original data presented in the study are openly available in the Open Science Framework (osf.io/eq5js).

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
