# Peer review of "Behavioral Emotion Regulation Strategies and Symptoms of Psychological Distress Among Turkish University Students"

_behavsci, 2024, doi:10.3390/bs15010006_

Round 1
Reviewer 1 Report
Comments and Suggestions for Authors
The study aims to investigate the relationship between behavioral emotion regulation strategies and symptoms of psychological distress, including depression, anxiety, and stress, among Turkish university students. This study contributes to the understanding of how specific emotion regulation behaviors may influence mental health outcomes in a student population. However, there are some revision requests that may need to be addressed.
1. In the introduction section, the authors should articulate the specific research hypotheses of this study based on relevant theoretical frameworks. This will set the stage for the investigation and guide the readers through the rationale behind the research objectives.
2. Descriptive statistics are mentioned, but it would be helpful to include the actual demographic characteristics of the sample in the results section to give readers a clearer picture of the participant profile.
3. In the discussion section, it may be insufficient to only consider cultural differences when exploring the unexpected findings (e.g., the positive association between seeking social support and psychological distress). The authors could draw upon relevant theories account for the heterogeneity of the population to provide a more in-depth explanation.
Author Response
Comment#1. The study aims to investigate the relationship between behavioral emotion regulation strategies and symptoms of psychological distress, including depression, anxiety, and stress, among Turkish university students. This study contributes to the understanding of how specific emotion regulation behaviors may influence mental health outcomes in a student population. However, there are some revision requests that may need to be addressed.
Thanks for this positive comment. There is nothing to change for this comment.
Comment#2. In the introduction section, the authors should articulate the specific research hypotheses of this study based on relevant theoretical frameworks. This will set the stage for the investigation and guide the readers through the rationale behind the research objectives.
|
Thanks for this helpful comment. We added following to Results section in a table. We only collected gender age and grade level of participants. We also made changes in Participants section after these additions to be consistent with each other. Our additions are highlighted using yellow fonts. 3.1. Sociodemographic Characteristics of the Participants Table 1 presents the sociodemographic characteristics of the participants including gender, age, and grade level distribution of the participants. As shown in Table 1, there were 477 female (75.4%) and 156 male (24.6%) students in the study sample. The age distribution of the participants ranged from 18 to 28 years, with a mean age of 21.28 years (SD = 1.77). Regarding academic grade level distribution, fourth-year students constituted the largest group (n = 209; 33%), followed by third-year students (n = 158; 25%). The remaining participants were distributed between second-year (n = 140; 22.1%) and first-year students (n = 126; 19.9%), indicating a relatively balanced representation across academic years with a slight greater representation of upper-level students. Comment#4. In the discussion section, it may be insufficient to only consider cultural differences when exploring the unexpected findings (e.g., the positive association between seeking social support and psychological distress). The authors could draw upon relevant theories account for the heterogeneity of the population to provide a more in-depth explanation. Thanks for these helpful comments. We added following to Discussion.
The unexpected positive association between seeking social support and symptoms of depression, anxiety and stress can be explain from the perspectives of Gross’s emotion regulation theory [20,21], Lazarus and Folkman’s stress and coping theory [56], and stress-mobilizing hypothesis [57] as well as findings from previous studies [54,55]. Gross’s emotion regulation theory [20,21] suggest that seeking social support can be viewed as an antecedent-focused strategy, where individuals proactively attempt to alter the emotional impact of a stressor by engaging others. While this strategy often has adaptive potential, its effectiveness depends on whether the social support aligns with the individual’s needs and expectations. If the support received is mismatched (e.g., emotional support when instrumental help is required) the recipient needs or if the interaction between the support recipient and provider intensifies nega-tive emotions, the strategy may fail to regulate distress effectively. Moreover, Lazarus and Folkman’s stress and coping theory [56] as well as previous studies [54,55] high-lights the individual and contextual factors that may influence the effectiveness of seeking social support as a coping mechanism including the relationship between the support recipient and provider, the source of support, the type of support, perceived versus received support, nature and severity of the stressors, and the current level of psychological distress of the support recipients. For example, students experiencing higher levels of psychological distress may have a reduced ability to perceive support as helpful, even if it is offered. Additionally, the nature and severity of the stressors, combined with the support provider's characteristics such as emotional availability or empathy, can influence the expected outcome for students. These complex dynamics can create a paradoxical situation where increased support-seeking behaviors may actually function as a stress-mobilizing mechanism [57]. Specifically, highly distressed students may intensify their attempts to access their social network, but unsuccessful relief efforts can generate a self-perpetuating cycle of unfulfilled needs and escalating distress. In the context of Turkish university students, these theoretical perspectives suggest that in-creased seeking social support behaviors as a coping mechanism may predict higher levels of psychological distress. However, given the complex interplay of factors in-volved and inconsistent findings in current literature, further research is necessary to identify the specific mechanisms and contextual conditions in which seeking social support is effectively reduces psychological distress in this population. Comment #5. The quality of English does not limit my understanding of the research. Thanks for this positive comment. There is nothing to change for this comment. |

Reviewer 2 Report
Comments and Suggestions for Authors
Thank you for the opportunity to review this article. Overall, the manuscript is easy to follow. However, I have a few concerns and suggestions:
- After reading the article, I wonder what the theoretical and empirical rationale is for focusing specifically on college students in Turkey. What unique contextual, cultural, or social characteristics distinguish this group from their counterparts nationally and/or internationally? While the authors mention theoretical perspectives in a few places (e.g., page 2, line 98: “from a theoretical perspective, emotion regulation is intricately linked to coping [24]”), these theories are not fully discussed. It’s helpful to elaborate on how these theoretical perspectives support the study and its focus on this specific population.
- On page 4, lines 191-194, the authors state: “Methodological limitations in previous studies further complicated the interpretation of the results. Most previous studies relied on bivariate analyses to examine the relationship between behavioral emotion regulation skills and symptoms of depression, anxiety, and stress.” However, there are comprehensive meta-analyses and reviews that have examined these relationships more rigorously, such as Compas et al. (2017, which the authors have cited [24] and Aldao et al. (2010). Acknowledging this body of prior research and clarifying how this study builds on or diverges from these established findings are necessary.
- I have concerns regarding the mandatory response format in Google Forms (Line 318). Forcing participants to answer all questions without the option to skip may raise ethical issues, particularly regarding the participants' right to withdraw or omit uncomfortable questions. This approach could violate the principle of voluntary participation. Could you please clarify how you ensured ethical compliance in the data collection process, especially in terms of participant consent and their ability to withdraw at any point?
- The manuscript does not clearly specify which covariates were adjusted for in the regression model, nor the rationale for selecting these covariates. It would be helpful to include a section that explains the theoretical or empirical reasoning behind the choice of covariates in your analysis.
- Finally, I am concerned about the generalizability of the findings. The sample appears restricted to a specific group of Turkish university students. To what extent can these findings be generalized to other populations, such as university students in different regions of Turkey or internationally? The limitation regarding generalizability (Line 573) acknowledges this issue but may not fully address readers' concerns about the broader applicability of the results.
Author Response
|
Thanks for this helpful comment. We revised limitations section. We added following to limitation section. We highlighted additions with yellow font. First, while we recruited a robust sample size of 633 students that exceeded our a priori power analyses requirements, our sample was limited to a convenience sample of university students from two institutions in the Central Black Sea Region and the Marmara Region of Türkiye. This non-probability sampling method restricts the generalizability of our findings to university students in other regions of Turkey and internationally, thereby limiting its external validity. The specific characteristics of our sample, drawn from these two distinct geographical regions, may not fully represent the Turkish university student population or university students living in other countries. To address these limitations, future researchers should employ larger, more diverse samples and multiple data collection methods. We recommend conducting multi-site studies that include university students from other regions of Türkiye as well as cross-cultural studies to test the generalizability of these findings. Comment#7. The quality of English does not limit my understanding of the research. Thanks for this positive comment. There is nothing to change for this comment.
|

Reviewer 3 Report
Comments and Suggestions for Authors
Review
Behavioral Emotion Regulation Strategies and Symptoms of 2 Psychological Distress among Turkish University Students
The present study examines 633 students in Turkey using a convenience sample. The aim is to examine relationships between emotion regulation strategies and psychological distress. The researchers performed descriptive statistics, correlations and multivariate regression. The paper in general is well written and describes the research topics well. However, there are some points which needs to be addressed:
Main Issues:
Study aims: It is unclear why the authors chose to focus their study on the variables emotion regulation strategies and anxiety and depression in Turkish students. There exists a strong evidence that emotion regulation strategies significantly impact mental health, with both adaptive and maladaptive strategies having distinct effects on mental health outcomes, such as anxiety and depressive symptoms. However, results may be particularly relevant for students in Turkey. The authors may describe therefore more clearly what impact the results in the particular sample for students have, and should also point our clearly the limitation, that no representative sample for Turkish students have been drawn.
Statistics: Oftentimes there exists intercorrelation between variables which can increase computational complexities tremendously. For example, to include all covariates in a multiple regression model will likely lead to severe multicollinearity issues. Therefore, recalculation using only one emotion regulation strategy in the model but controlled by age and gender is recommended. The result that seeking social support is positively associated with higher levels of depression, anxiety, and stress could be an artificial result as all emotion regulation strategies were entered simultaneously into the model.
Minor Problems:
Line 193: Sentence: Most previous studies relied on “bivariate analyses”. It is unclear what bivariate analyses are. Do you mean simple linear regressions or correlations based on two variables?
Author Response
Comment#1. Behavioral Emotion Regulation Strategies and Symptoms of Psychological Distress among Turkish University Students The present study examines 633 students in Turkey using a convenience sample. The aim is to examine relationships between emotion regulation strategies and psychological distress. The researchers performed descriptive statistics, correlations and multivariate regression. The paper in general is well written and describes the research topics well. However, there are some points which needs to be addressed
Thanks for this positive comment. There is nothing to change for this comment.
Study aims: It is unclear why the authors chose to focus their study on the variables emotion regulation strategies and anxiety and depression in Turkish students. There exists strong evidence that emotion regulation strategies significantly impact mental health, with both adaptive and maladaptive strategies having distinct effects on mental health outcomes, such as anxiety and depressive symptoms. However, results may be particularly relevant for students in Turkey. The authors may describe therefore more clearly what impact the results in the particular sample for students have.
|
Thanks for this helpful comment. We collected only gender, age, and grade level as sociodemographic variables in this study. Upon request of another reviewer and you, we conducted two separate multivariate multiple regression analyses using gender age behavioral emotion regulation strategies (Model 1), using gender grade level behavioral emotion regulation strategies (Model 2) as independent variables. Because grade level and age were strongly correlated (rs = .73). Results of this analyses reported as Supplementary material Table S1-Table S5. We also added these results end of this response letter. Moreover we added reasons for conducting multivariate multiple regression analyses to Introduction section. As seen in analyses, both sets of analyses yielded similar results, indicating that the relationships between behavioral emotion regulation strategies and symptoms of depression, anxiety, and stress were consistent across demographic factors. We also added following to Statistical Analysis Section. To assess the robustness of these relationships and control for potential sociodemographic effects, we performed two additional multivariate multiple regression analyses. Due to the strong correlation between grade level and age (rs = 0.73), we tested these sociodemographic variables in separate models to avoid multicollinearity. Model 1 included gender, age, and behavioral emotion regulation strategies as predictors, while Model 2 incorporated gender, grade level, and behavioral emotion regulation strategies. These supplementary analyses yielded results consistent with our primary findings, suggesting that the relationships between emotion regulation strategies and symptoms of depression, anxiety, and stress remained stable across demographic variables (see Supplementary Material, Tables S1-S5). Thus, we present our main findings without demographic covariates. We also added following to Introduction section. We highlighted additions with yellow font. Research has consistently demonstrated a strong link between emotion regulation and mental health [24,27]. Previous comprehensive reviews and meta-analyses such as those by Compas et al. [24] and Aldao et al. [27], have provided important insights into the relationships between various emotion regulation strategies and psychopathology. However, these studies have primarily focused on cognitive emotion regulation strategies, examining their associations with symptoms of depression, anxiety, and other mental health problems. Cognitive emotion regulation strategies involve complex mental processes that shape the emotional experience such as reappraisal and rumination [22,26,27]. In contrast, behavioral emotion regulation strategies allow individuals to directly modulate their emotional responses, particularly in the context of stressful situations [23]. However, behavioral emotion regulation strategies have received less attention in the literature compared to the cognitive strategies. We used a different approach by investigating the role of behavioral emotion regulation strategies, as conceptualized by Kraaij and Garnefski [23] in this study to provide a more nuanced understanding of how individuals actively manage their emotions through specific actions, rather than solely relying on cognitive processes. Methodological limitations in previous studies further complicated the interpretation of the results. Most previous studies relied on bivariate analyses, such as Pearson product-moment correlation coefficient analyses, to examine the relationships between behavioral emotion regulation strategies and the symptoms of depression, anxiety, and stress [28–32]. While these analyses offer valuable insights into the strength and direction of linear relationships between specific strategies and psycho-logical distress, they fail to account for the complex interplay among different behavioral strategies. However, linear multiple regression analyses or its extensions (e.g., multivariate multiple regression analysis) provide several advantages over correlation analysis [33–35]. First, linear multiple regression analysis allows researchers to identify the relative importance of each behavioral emotion regulation strategy in predicting psychological distress, rather than simple association. This information can be particularly useful for informing the development of targeted interventions that focus on strengthening the most influential behavioral emotion regulation strategies. Such knowledge can also enable mental health professionals to develop and tailor interventions more effectively, especially when working with university students. Second, lin-ear multiple regression analysis accounts for the potential overlap or shared variance among the different behavioral strategies, providing a clearer picture of the unique contributions of each behavioral strategy. This is critical because behavioral emotion regulation strategies often do not operate in isolation but interact in complex and interdependent ways. Finally, linear multiple regression analysis enables researchers to assess the overall model fit, indicating the collective explanatory power of the behavioral emotion regulation strategies in predicting symptoms of depression, anxiety, and stress [33–35]. The present study aims to determine the predictive role of behavioral emotion regulation strategies (seeking distraction, withdrawal, actively approaching, seeking social support, and ignoring) in explaining symptoms of depression, anxiety, and stress among Turkish university students. Comment#5. Line 193: Sentence: Most previous studies relied on “bivariate analyses”. It is unclear what bivariate analyses are. Do you mean simple linear regressions or correlations based on two variables? Thanks for this helpful comment. We added following to Introduction section for clarification. We highlighted additions with yellow font. Methodological limitations in previous studies further complicated the interpretation of the results. Most previous studies relied on bivariate analyses, such as Pearson product-moment correlation coefficient analyses, to examine the relationships between behavioral emotion regulation strategies and the symptoms of depression, anxiety, and stress [28–32]. Comment#6. The quality of English does not limit my understanding of the research. Thanks for this positive comment. There is nothing to change for this comment. |
